# Numerical Study of Ti6Al4V Alloy Tube Heated by Super-Frequency Induction Heating

**DOI:** 10.3390/ma16113938

**Published:** 2023-05-24

**Authors:** Cheng Liu, Jingtao Han, Ruilong Lu, Jiawei Liu, Xiaoyan Ma

**Affiliations:** 1School of Materials Science and Engineering, University of Science and Technology Beijing, Beijing 100083, China; 2Guangzhou Sino Precision Steel Tube Industry Research Institute Co., Ltd., Guangzhou 511300, China

**Keywords:** numerical simulation, induction heating, electromagnetic field, thermal field

## Abstract

Ti6Al4V alloys have a narrow processing window, which complicates temperature control, especially during large-scale production. Therefore, a numerical simulation and experimental study on the ultrasonic induction heating process of a Ti6Al4V titanium alloy tube were conducted to obtain stable heating. The electromagnetic and thermal fields in the process of ultrasonic frequency induction heating were calculated. The effects of the current frequency and current value on the thermal and current fields were numerically analyzed. The increase in current frequency enhances the skin and edge effects, but heat permeability was achieved in the super audio frequency range, and the temperature difference between the interior and exterior of the tube was less than 1%. An increase in the applied current value and current frequency caused an increase in the tube’s temperature, but the influence of current was more prominent. Therefore, the influence of stepwise feeding, reciprocating motion, and stepwise feeding superimposed motion on the heating temperature field of the tube blank was evaluated. The coil reciprocating with the roll can maintain the temperature of the tube within the target temperature range during the deformation stage. The simulation results were validated experimentally, which demonstrated good agreement between the results. The numerical simulation method can be used to monitor the temperature distribution of Ti6Al4V alloy tubes during the super-frequency induction heating process. This is an economical and effective tool for predicting the induction heating process of Ti6Al4V alloy tubes. Moreover, online induction heating in the form of reciprocating motion is a feasible strategy for processing Ti6Al4V alloy tubes.

## 1. Introduction

Titanium and titanium alloys are extensively used in the aerospace, medical, and other fields owing to their high strength, light weight, corrosion resistance, and excellent high-temperature performance [1,2,3]. The Ti6Al4V alloy is a typical α-β dual-phase alloy. The common fabrication methods for Ti6Al4V tubes include hot extrusion, cross rolling, and hot rolling [4,5,6], which are mostly used to produce large-diameter and thick-walled tube blanks [7]. Additionally, because of the narrow processing window of the Ti6Al4V alloy [8], temperature control is challenging, and large-scale production such as steel has not been achieved. In the context of materials processing, the term “processing window” refers to the range of parameters within which a material can be successfully manipulated or fabricated to achieve the desired properties. These parameters may include factors such as the temperature, strain rate, cooling rate, and deformation conditions. Ti6Al4V undergoes several phase transformations during processing, primarily involving the transformation between the α and β phases. The phase transformations are highly temperature-dependent and can significantly influence the material’s mechanical properties. Precise control of the temperature is necessary to achieve the desired phase balance and avoid the formation of undesirable phases. In subsequent applications, the processing of small- and medium-diameter Ti6Al4V alloy tubes is primarily performed by cold rolling [9]. Owing to the poor room temperature plasticity and large deformation resistance of the Ti6Al4V alloy, the equipment requirements are very high, and the cold deformation rate is usually less than 15% [10]. Therefore, it is commonly processed by multi-pass cold rolling combined with intermediate heat treatment, which is complex.

Currently, reports on the formation of Ti6Al4V alloys mainly focus on the evolution of the microstructure and texture after deformation, and the influence of the heat treatment process on the microstructural and textural properties [11,12,13,14]. Although some new processing methods for small- and medium-sized Ti6Al4V alloy tubes have been explored, there are still some shortcomings. For example, Yuan et al. [15,16] prepared small-diameter, thin-walled Ti6Al4V alloy tubes for aviation hydraulic systems using cold rotary forging. The maximum deformation was 51%, but distinct cracks were observed when the deformation exceeded 55%. However, the subsequent application should be combined with heat treatment to eliminate residual stress, which has certain limitations.

Therefore, it is necessary to continuously explore the formation of small- and medium-sized Ti6Al4V alloy tubes and to develop new processes to solve their processing complexities.

In this study, a temperature-controlled periodic rolling process based on a cold rolling mill is developed. The heating method involves induction heating, and the heating coil synchronously reciprocates with the roll, which circumvents the severe temperature reduction during the tube step feeding process. The core of this process is to control the temperature during hot processing. Researchers have always aimed to obtain a more accurate heating temperature distribution during hot processing [17,18,19,20,21,22] because temperature control changes the microstructure and texture of titanium alloys and alters their mechanical behavior [23,24,25,26]. Electromagnetic induction heating technology has the advantages of high performance and environmental protection. Moreover, owing to the continuous development of computer digital technology, accurate electromagnetic heating has attracted increasing attention [27,28,29,30]. Because the induction heating process is complex, involving electromagnetic, temperature, and force fields, numerical simulations can directly and accurately study the physical field compared with the experimental method [31,32,33,34].

Studying small and medium-sized Ti6Al4V alloy tube induction heating using numerical simulations presents several challenges. Induction heating relies on the interaction of electromagnetic fields and the conductive material. Simulating the complex electromagnetic phenomena accurately requires solving Maxwell’s equations coupled with heat transfer and solid mechanics equations. Handling the electromagnetic boundary conditions and accurately predicting the eddy currents and their distribution can be challenging. What is more, induction heating involves time-dependent processes, where the heat is induced and gradually propagates through the material. Simulating the transient behavior of heating, temperature distribution, and thermal gradients within the tube requires solving time-dependent equations, which can significantly increase the computational cost. Accurately capturing the thermal and electromagnetic properties of Ti6Al4V alloy is essential for reliable simulations. Achieving convergence in numerical simulations is crucial to ensure accurate results. The simulation parameters such as mesh size, time step, and solver settings need to be carefully chosen and validated to ensure convergence. Complex geometries and time-dependent phenomena can make convergence more difficult to achieve, requiring thorough verification and validation studies. Validating the numerical simulations against experimental data is essential to establish confidence in the results. However, to obtain experimental data for small and medium-sized alloy tubes, calibration of the simulation parameters and models to match experimental observations becomes a critical step. Therefore, this study uses a combination of numerical simulations and experimental research to focus on the electromagnetic and temperature fields in the super-frequency induction heating process to understand these fields in Ti6Al4V alloy tubes under these conditions. Combined with the motion characteristics of the cold rolling mill, a heating form that can realize stable temperature control in the rolling process is evaluated by changing the motion form of the coil. The main aim of this study is to achieve stable temperature control during the rolling process and provide new possibilities for the processing of Ti6Al4V alloy tubes.

## 2. Numerical Simulations

### 2.1. Numerical Model Description

The working principle of the online super-frequency induction heating pilger rolling equipment developed in-house is illustrated in Figure 1. This study focused on the induction heating process. Therefore, only a finite element model of the induction heating part was provided, and the hot rolling model was omitted. To improve computational efficiency, the model was reasonably simplified and enclosed within a closed box, representing the far field. The spiral coil was simplified as a closed ring positioned alongside the tube blank. Since the coils are arranged parallel to the length of the tube and are coaxial with the tube blank, a two-dimensional model (Figure 2a) is selected by taking a cross-section, simplifying the original three-dimensional model for calculations. This simplification greatly enhances solution efficiency. The geometric models and the contact conditions of the models are shown in Figure 2b where the workpiece and coil are surrounded by inner air, inner air is surrounded by outer air, and outermost air is the far field. The contact was set to paste, and the initial temperature of all components was 20 °C. The magnetic insulation condition is applied in the far field; the edge condition of the contact between the workpiece and coil surface and the near-air field is continuous.

Grid dependence analysis reveals that, despite occupying the largest area, the outer air has the least impact and can be considered least important. The coil model, tube blank model, and the inner air surrounding the coil are critical factors influencing simulation results and should be prioritized in the modeling process. Grids should have a regular shape and smaller size to ensure accuracy. However, after a certain level of refinement, although the number of iterations increases significantly, the differences in temperature field display results become less pronounced. Therefore, the grid division presented in Figure 3 was employed. Figure 3a shows the overall situation, and Figure 3b shows the local amplification near the coil and the tube. The specific grid types and parameters are listed in Table 1.

The workpiece material studied was Ti6Al4V with an outer diameter of 45 mm and a wall thickness of 3 mm. The workpiece was numerically studied in three forms: fixed, stepwise feed at a certain speed, and superimposed motion. Because the induction heating coil should be fixed in the simulation, the superposed motion applied a reciprocating motion and stepwise feed to the workpiece according to the principle of relative motion. The stepwise feed time interval was 1 s, and the feed was 3 mm. The reciprocating motion was obtained using the SolidWorks 2018 software to construct a solid model for motion analysis based on the kinematic characteristics of the offset crank connecting rod slider–crank mechanism. The simulation conditions included the connecting rod length (1460 mm), crank radius (250 mm), offset (170 mm), and rotational speed (60 rpm). The workpiece motion curves are shown in Figure 4 (first 10 s). Figure 4a shows the feeding curve of the tube, Figure 4b shows the motion curve of the rolling mill; when the two motions are superimposed on the coil, time stroke curve is as shown in Figure 4c. The coil material was a cylindrical copper coil with an outer diameter of 10 mm and an inner diameter of 7 mm. There were six coil turns, and the overall coil diameter was 60 mm. The physical parameters of the copper coil and air are listed in Table 2. In this study, the thermophysical and physical properties of the heated workpiece were defined in advance. The material properties were calculated using the JmatPro software to obtain the relationship between electrical conductivity, thermal conductivity, specific heat, and temperature, as shown in Figure 5a–c. M. Boivineau et al. [35] conducted a study on the thermophysical properties of solid and liquid Ti6Al4V alloys using two different experimental devices. The conductivity, thermal conductivity, and specific heat presented in their article are basically consistent with the trend of electromagnetic and thermal parameters with temperature simulated by JmatPro 9.0 software in this paper. Therefore, we believe that the physical parameters simulated by JmatPro software can be used in subsequent calculations. The calculation results were imported into the Simufact Forming 16.0 software for simulation.

### 2.2. Electromagnetic Fields

Maxwell’s equations were used to calculate the electromagnetic fields. This analysis is called the magnetodynamic simulation. Maxwell’s equations are written as follows:(1)$$\left\{\begin{array}{c} \nabla \cdot \vec{D}=\rho \\ \nabla \cdot \vec{B}=0\\ \nabla \times \vec{H}=\vec{J}+\frac{\partial \vec{D}}{\partial t}\\ \nabla \times \vec{E}=-\frac{\partial \vec{B}}{\partial t} \end{array}\right.,$$

When there is no free charge in the analysis domain, the equations can be simplified as:(2)$$\left\{\begin{array}{c} \nabla \cdot \vec{D}=0\\ \nabla \cdot \vec{B}=0\\ \nabla \times \vec{H}=\vec{J}\\ \nabla \times \vec{E}=-\frac{\partial \vec{B}}{\partial t} \end{array}\right.,$$

In practical calculation, vector magnetic potential and scalar potential are usually introduced to reduce the amount of calculation:(3)$$\left\{\begin{array}{c} \vec{B}=\nabla \times \vec{A}\\ \vec{E}=-\frac{\partial \vec{A}}{\partial t}-\nabla \cdot \varphi \end{array}\right.,$$

Then, according to the relationship between magnetic induction intensity vector, magnetic field intensity vector, current density vector, and electric field intensity vector, the electromagnetic field control equation can be obtained. Table 3 summarizes the nomenclature of all the parameters used for the analysis.

### 2.3. Temperature Fields

The temperature field of the induction heating process can be described by a classical transient heat transfer equation as follows:(4)$$\rho^{\prime }c\frac{\partial T}{\partial t}-\lambda \;div\left(grad\;T\right)=Q,$$
or
(5)$$\rho^{\prime }c\frac{\partial T}{\partial t}-\lambda \nabla^{2}T=Q,$$
and because
(6)$$Q=\frac{1}{\sigma }{\left|\vec{J}\right|}^{2},$$
then the induced current density and conductivity are substituted, and combined with the heat transfer conditions, the solution can be solved.

In the process of heating and cooling, the heat transfer mode mainly includes heat conduction, heat convection, and heat radiation. Among them, heat conduction is mainly the heat transfer between different temperature regions inside the tube billet, while heat convection and heat radiation are the heat transfer from the inner and outer surfaces of the tube billet to the surrounding air environment. The boundary conditions can be described as follows:(7)$$Q_{c}=Ah\left(T_{S}-T_{E}\right),$$
(8)$$Q_{r}=\sigma_{\mathrm{sb}}\varepsilon A_{S}\left(T_{S}^{4}-T_{E}^{4}\right),$$
where *Q_c_* is the convicted power, *A* is the area, *T_S_* is the workpiece temperature, *T_E_* is the ambient temperature, *Q_r_* is the radiated power.

### 2.4. Experimental Setup

The heating and temperature measurement experiments were divided into two types. First, the workpiece was relatively fixed with the coil. Second, the workpiece was fed in a stepwise manner, while the coil was reciprocated in the stroke.

When the workpiece and coil were fixed, the heating conditions were as follows: the heating time was 15 s; when the current frequency was fixed at 10 kHz, the current values were 100, 200, 300, and 400 A; when the current value was fixed at 100 A, the current frequencies were 20, 30, and 40 kHz, which is a total of seven groups, and each group was repeated three times. After heating was completed, the coil was immediately removed, and the temperature change under different current values and current frequencies was measured using an infrared thermal imager (FLIR Systems, Inc., Portland, OR, USA).

The workpiece was fed in a stepwise manner. The feed interval was 1 s, the feed distance was 3 mm each time, the coil performs reciprocating motion in the stroke at a frequency of one time per second. The process temperature was measured using an infrared thermal imager (FLIR Systems, Inc., Portland, OR, USA). The temperature measurement range was from −20 to 2000 °C, the wavelength range was ~7.5–14 μm, the measurement accuracy was ±3%, and the emissivity was 0.3 [36]. Additionally, the measurement distance of autofocus and laser positioning was 1.2 m.

## 3. Results and Discussion

### 3.1. Electromagnetic and Thermal Analysis

Figure 6a–d show the distribution of the magnetic potential around the coil when the current frequency was 10 kHz and the current was 100, 200, 300, and 400 A. The magnetic potential describes the magnetic field penetrating the air and nearby workpiece. The magnetic potential distribution diverges from the coil at the center of the four-circle ellipse. The highest intensity appeared near the coil, and the side near the workpiece had a higher density. However, the workpiece had a certain limit on the magnetic field. The shape of magnetic potential did not change significantly with increasing current, but the corresponding color value increased.

Figure 7a–d show the electric current density in the workpiece at different current values when the current frequency was 10 kHz. The electric current density represents the area in which the electrical current is concentrated. As shown in Figure 7, owing to the skin effect, the electrical current was mainly concentrated on the surface of the workpiece, and the inner current density gradually decreased. With an increase in the current value, the electrical current density increased continuously, which resulted in an increase in the heat flux and temperature in this region. Figure 8a–c and d show the temperature field corresponding to the different current values when the heating time was 15 s. The maximum heating temperature increased from 140.41 °C at 100 A to 1296.81 °C at 400 A. According to Joule–Lenz’s law, the Joule heat generated by the eddy current is proportional to the square of the current. The increase in the input alternating current causes the increase in the alternating magnetic field, which in turn causes the increase in the induced electromotive force of the metal conductor placed in the alternating magnetic field, thereby generating more Joule heat and causing the workpiece temperature to rise rapidly.

Based on “skin effect”, frequency has a significant effect on the penetration depth of heating. Therefore, it is necessary to evaluate the electromagnetic field at different frequencies. Figure 9a–c and d show the change in magnetic potential at 10, 20, 30, and 40 kHz when the current was 100 A. With the increase in frequency, the magnetic field became more concentrated on the side near the workpiece. Moreover, at a frequency larger than 30 kHz, the magnetic potential at the top edge of the tube was also more concentrated and had the shape of an “ear”.

Figure 10a–d show the electric current density in the workpiece at 10, 20, 30, and 40 kHz when the current was 100 A. It can be seen that, as the current frequency increased, the current density increased continuously, and it was more concentrated on the outer surface of the workpiece. Furthermore, the current density at the top edge of the tube increased significantly, which may be owing to the edge effect [37]. An increase in the current density inevitably causes an increase in temperature. Figure 11a–d are the temperature fields at 10, 20, 30 and 40 kHz when the current is 100 A. It can be seen that, as the current frequency increased, the top area near the edge of the tube gradually became the highest-temperature region. The excessive temperature difference between the top edge of the workpiece and other parts inevitably leads to the difference in microstructure and properties after forming. This process is expected to achieve uniform and permeable heat of the workpiece, which is different from the high-frequency welding process so that the temperature is concentrated at the edge of the plate. In order to avoid the waste of materials, this difference should be minimized.

To understand the temperature difference in the workpiece along the thickness direction. The preset point temperature was recorded in the wall thickness direction, and the selected position of the point is shown in Figure 12a. Figure 12b shows the temperature of the selected point in the wall thickness direction under different current values and current frequencies when the heating time was 15 s. On a large scale, the temperature distribution in the wall thickness direction was almost linear. As shown in Figure 12d,e, after the scale was reduced, the temperature change exhibited a gradual increase from the interior to the exterior of the workpiece, and finally, a slight decrease near the outermost layer. The temperature reduction in the outermost layer may be related to heat dissipation from the workpiece to the air. Although there are some differences in the temperatures of the inner and outer layers, the temperature difference does not exceed 1%, indicating that the wall thickness direction achieved good heat permeability at current values of 100–400 A and current frequencies of 10–40 kHz. Figure 12c shows the temperature–time curve of point 69 at different current values and current frequencies. Under the same proportion of changes, the influence of the applied current on the rate of temperature increase was more obvious than that of the current frequency. The effect of current value and frequency value on workpiece temperature agrees with the results reported by L Lu et al. [36], while their research is concerned with the induction heating melting of titanium wire.

Based on the influence of the current value and current frequency on the workpiece temperature, when the target temperature was set, the workpiece reached the target temperature within a specific time by adjusting the current value and current frequency. This process should be based on the current value adjustment supplemented by the current frequency adjustment. Notably, current adjustment can prevent the high temperature at the top of the workpiece caused by high frequencies. Thus, to meet the target temperature range, the current value can be appropriately increased, and the current frequency can be reduced, which is conducive to achieving heat equalization in the wall thickness direction.

### 3.2. The Influence of Motion on Heating

After understanding the variations in the electromagnetic and thermal fields when the coil and workpiece were relatively fixed, it is necessary to further explore the heating changes caused by motion. Figure 13a–f show the temperature cloud at different times when the workpiece is fed in a stepped manner at a current value of 300 A and a current frequency of 20 kHz. The feed length was 3 mm and the feed interval was 1 s. The motion curve is shown in Figure 4a (first 10 s).

As shown in Figure 13, as the workpiece entered the coil area, the temperature gradually increased. After entering the coil completely, the temperature reached the maximum value, the workpiece left the coil, and the temperature gradually decreased. During the entire process, a distinct temperature gradient was observed in the feeding direction of the workpiece temperature, which is due to the difference in the heating time caused by the order of the workpiece entering the coil. Stepwise feeding leads to a long heating time for the part entering the coil first and a short heating time for the part entering the coil later. The entire process was progressively repetitive. It should be noted that in the actual online induction heating rolling, the tube length can reach 4 m. The simulation in this paper only gives a length of tens of centimeters because most of the tubes are always in a stable online heating state except when the tube just enters and leaves the coil.

Because the temperature of the workpiece decreases rapidly when it is outside of the coil, the temperature required for rolling cannot be achieved when it reaches the roll. To prevent severe temperature reduction in the workpiece during the stepwise feeding process, it is necessary to improve the heating method such that the workpiece is constantly heated during the main deformation stage.

To solve this problem, the coil and roll were connected as a whole, such that the coil and roll reciprocate motion synchronously. The stroke of the roller was the heating stroke of the coil. Because the coil must remain fixed in the simulation, according to the principle of relative motion, reciprocating and stepwise motions were applied to the tube blank. The motion curve is shown in Figure 4c. Figure 14 shows the relative position change of the workpiece and coil at different times when superimposed motion was applied to the workpiece. In this process, the change in diameter caused by rolling was not considered. Owing to the repeatability of the process, a tube length of 130 mm was used in the simulation to minimize the calculation time.

Figure 15a–e show the temperature cloud diagram of the workpiece at different feeding times at a current of 400 A and frequency of 30 kHz. The temperature of the workpiece did not decrease significantly during the entire feeding process, and the temperature of the workpiece became uniform as the workpiece was fed.

To more clearly reflect the temperature changes of each part of the workpiece, the temperatures of the points shown in Figure 16a were monitored to detect the temperature changes at each point during the movement.

As shown in Figure 16b, with the exception of the temperature at points 309 and 304 at the top of the workpiece, which is slightly higher than that of the subsequent points, the temperature change trend of each point was similar, exhibiting a stable state after heating up to a certain temperature. When the workpiece was about to exceed the coil stroke, the temperature began to decrease. The reason for the slightly higher temperature at the top of the workpiece was identical to that when the coil was fixed, which was the edge effect. Point 299 can be used as a demarcation point for entering a stable heating state. This point represents the change rule of the subsequent points, and the maximum temperature was stable at approximately 925 °C. The heat transfer from the coil to the workpiece and the heat dissipation of the workpiece to the exterior reached equilibrium.

As shown in Figure 14, the distance from the beginning of the workpiece entering the coil to entering the roll was L1 + L2. Because the workpiece is accompanied by rotation during the feeding process, to achieve smooth rotation, the roll was not in contact with the workpiece at the rear limit position; that is, it does not immediately begin to deform after entering the roll but continues to move forward the L3 distance, and then the workpiece begins to roll. At this time, the feed distance was L1 + L2 + L3 = 85 + 120 + 90 = 295 mm, and it took approximately 95 s to feed at a rate of 3 mm/s.

The times corresponding to the green vertical dashed lines in Figure 16b were 95 and 180 s, which represent the time when the workpiece begins to deform and the completion time of the main deformation stage, respectively. As shown in Figure 16a, the distance between points 299 and 309 was 30 mm. Because the feed rate was 3 mm/s, the time difference between these points was approximately 10 s. Therefore, the times corresponding to the blue vertical dashed lines were 105 and 190 s, which represent the time at the start of deformation and the completion of the main deformation stage at point 299, respectively. As shown in Figure 16b, the temperature was maintained at approximately 925 °C during this period, which realizes the effect of stable online heating.

Compared with the case when the coil is fixed, the reciprocating motion of the coil extends the heating distance of the tube. Therefore, the tube is always heated during the coil stroke. However, the heating time of each point in the same cycle becomes shorter, so more heating time is required. From Equations (5)–(7), it can be seen that the acquisition and dissipation of heat determine the final temperature state of the tube blank. With the increase in temperature, the dissipation of heat accelerates. If the heating efficiency does not change much, the increase in heat dissipation will inevitably make the tube blank temperature reach an equilibrium state at a certain value.

It should be pointed out that the different physical properties of the material and the change in the feeding speed will inevitably lead to the change in the induction heating temperature of the workpiece. In this paper, only the case of the feeding speed of 3 mm/s of the Ti6Al4V tube was studied. In the future, the workpiece heating under other materials and different feeding conditions can be further explored.

### 3.3. Experiment Validation

To verify the accuracy of the induction heating temperature field of the Ti6Al4V alloy tube in the simulation, its induction heating temperature was measured under different current values and current frequencies. Figure 17a shows the temperature of the Ti6Al4V alloy tube heated for 15 s under different currents at a frequency of 10 kHz. Figure 17b shows the temperature of the Ti6Al4V alloy tube heated at different current frequencies for 15 s at a current value of 100 A. In the process of induction heating, the workpiece temperature increased with an increase in the current value and current frequency, and the influence of the current value on the temperature was more significant than that of the current frequency. The simulated temperature was consistent with the experimental data.

Figure 18a shows that when the superimposed motion was 140 s, the experimental and simulated temperature fields were compared. Based on the color, the temperature distribution of the workpiece in the length direction was uniform, and the temperature labels in the simulation and experiment were similar. Points in the length direction of the workpiece were selected, the temperature of the points was measured, and the temperature distribution of the workpiece in the length direction was plotted, as shown in Figure 18b. The total distance between the points was approximately 90 mm. The temperature distribution of the workpiece in the length direction was not significantly different, and the measured temperature fluctuation was slightly larger than the numerical simulation. The simulated temperature fluctuated at approximately 925 °C and the experimentally measured value fluctuated at approximately 938 °C; however, in general, the temperature difference was insignificant. The temperature difference between the two values was less than 20 °C, and the simulated values were consistent with the experimentally measured values. In plastic forming, temperature is a crucial parameter. The increase in temperature can soften the material, thereby reducing the forming force and making the metal easy to deform. Since this paper mainly focuses on the induction heating of an Ti6Al4V alloy tube, the subsequent hot pilger rolling deformation is not discussed. However, in the subsequent actual processing, we found that the deformation rate of the tube in the single-pass rolling deformation can reach 70%, which is much larger than the deformation rate of the cold rolling of 15%. This provides the possibility to replace the processing scheme of “multi-pass cold deformation plus multi-pass intermediate heat treatment”.

## 4. Conclusions

In this study, a numerical simulation and experimental study of ultrasonic frequency induction heating of Ti6Al4V alloy tubes were conducted. A finite element model of electromagnetic–thermal coupling was developed. The electromagnetic and temperature fields at different current values and current frequencies were evaluated, and the temperature field under different motion states was analyzed. The experimental results were in good agreement with the numerical simulation results. The main conclusions are as follows.

(1)The temperature of the Ti6Al4V alloy tube increased with an increase in the current value and current frequency, but the influence of the current value was more significant.(2)An increase in the current frequency enhanced the edge effect, and the temperature at the top of the tube was higher than that of the other parts. At the same target temperature, a large current value and small current frequency were considered more suitable than a small current value.(3)When the current frequency was within the range of 10–40 kHz, the Ti6Al4V alloy tube with a wall thickness of 3 mm achieved heat permeability, and the temperature difference in the wall thickness direction did not exceed 1%.(4)When the coil undergoes reciprocating motion with the rolling tube at a current value of 400 A and current frequency of 30 kHz, the temperature of the Ti6Al4V alloy tube was relatively stable between 925 and 945 °C, which realizes online stable heating of the Ti6Al4V alloy tube and presents a new strategy for titanium alloy tube processing.

This study provides a reference for the temperature field analysis during the super-frequency induction heating of titanium alloy tubes under fixed and reciprocating motions. In the future, the workpiece heating under other materials and different feeding conditions can be further explored.

## Figures and Tables

**Figure 1 materials-16-03938-f001:**
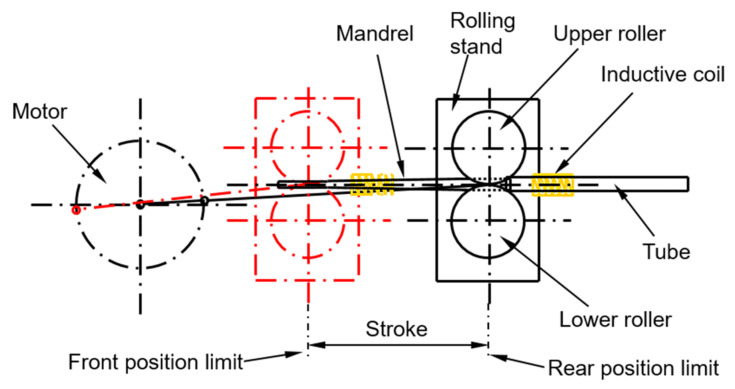
Schematic of online induction heating pilger hot rolling equipment.

**Figure 2 materials-16-03938-f002:**
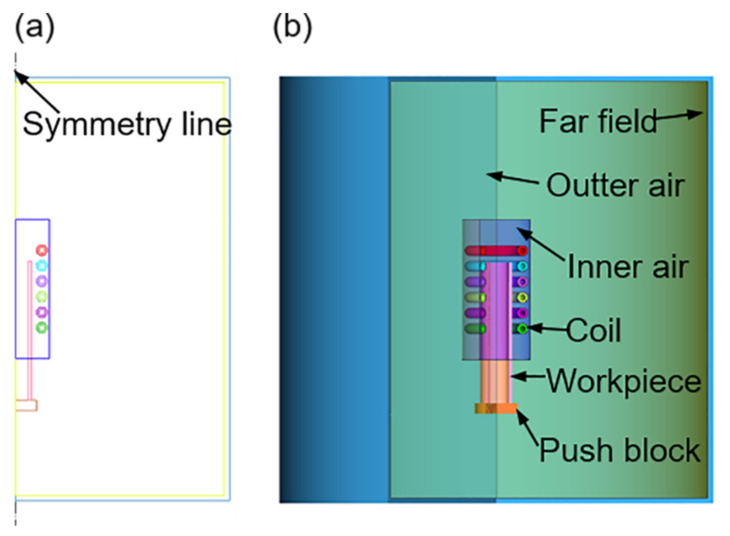
Geometric models: (**a**) Two-dimensional and (**b**) three-dimensional.

**Figure 3 materials-16-03938-f003:**
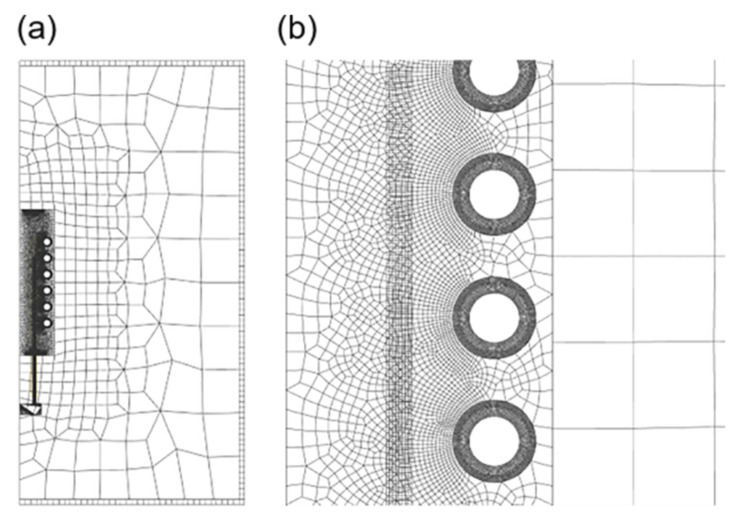
FE mesh plot of the model: (**a**) Overall and (**b**) local amplification.

**Figure 4 materials-16-03938-f004:**
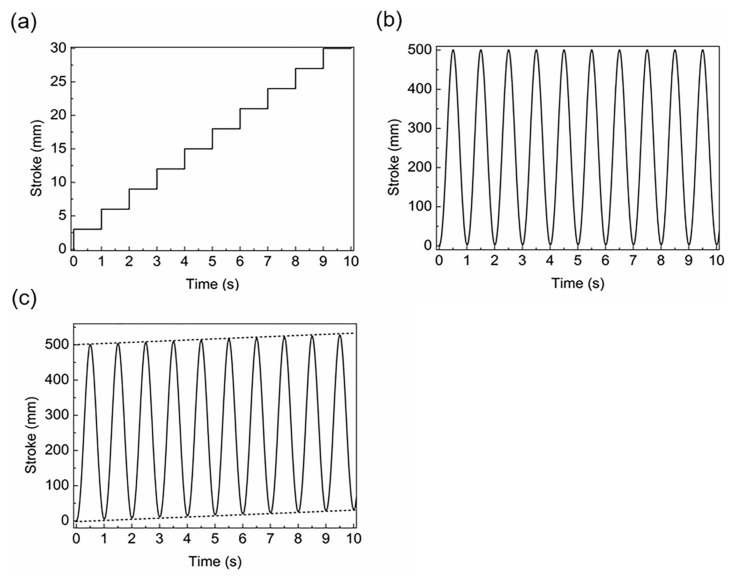
Time stroke curves: (**a**) feeding, (**b**) reciprocating motion; (**c**) superposed motion.

**Figure 5 materials-16-03938-f005:**
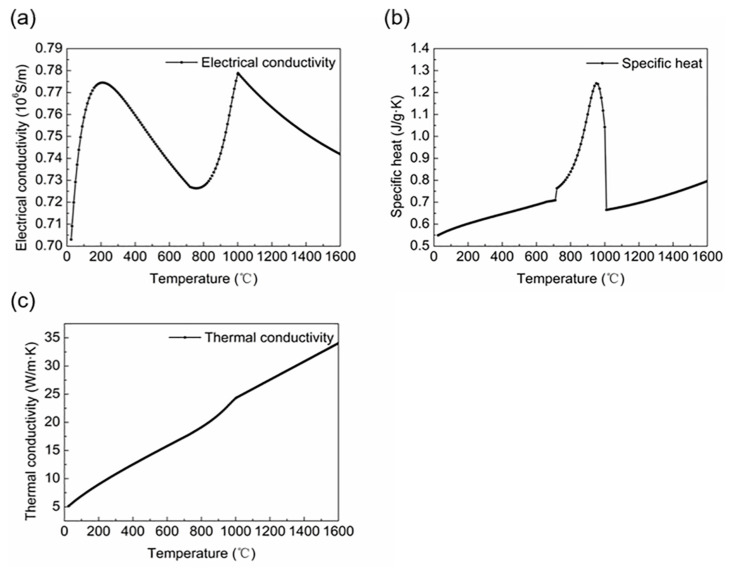
Physical properties of the material as a function of temperature: (**a**) electrical conductivity; (**b**) specific heat; (**c**) thermal conductivity.

**Figure 6 materials-16-03938-f006:**
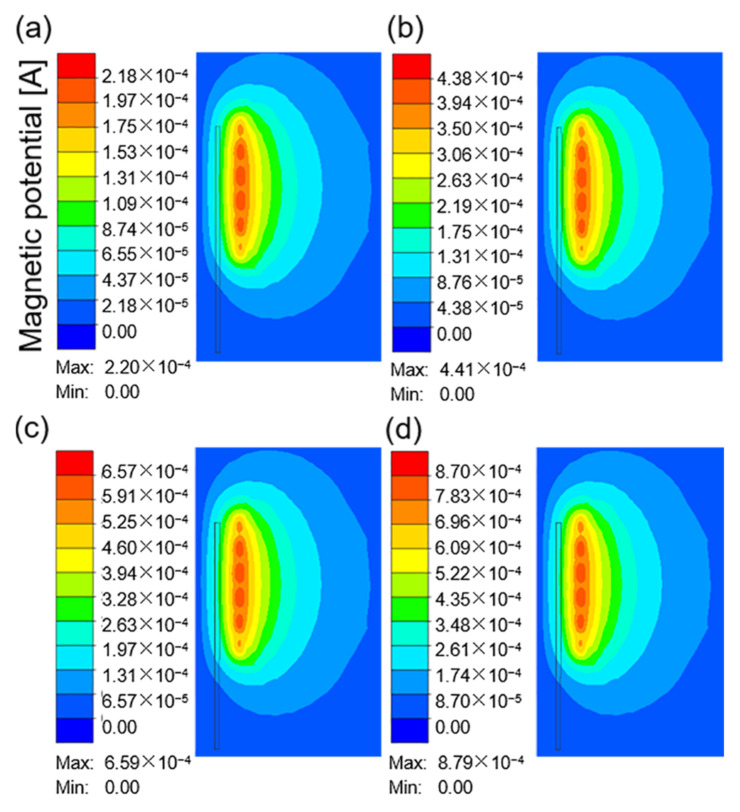
The magnetic potential around the coil at a current frequency of 10 kHz and different current values: (**a**) 100 A; (**b**) 200 A; (**c**) 300 A; (**d**) 400 A.

**Figure 7 materials-16-03938-f007:**
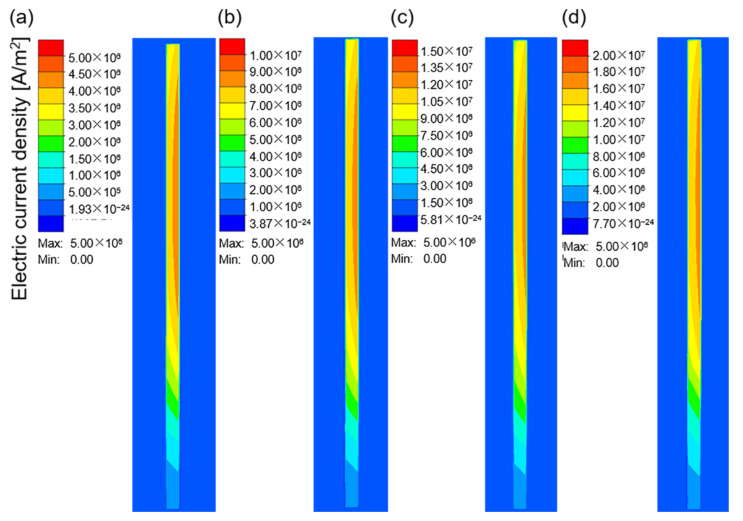
The electric current density in the workpiece at a the current frequency of 10 kHz and different current values: (**a**) 100 A; (**b**) 200 A; (**c**) 300 A; (**d**) 400 A.

**Figure 8 materials-16-03938-f008:**
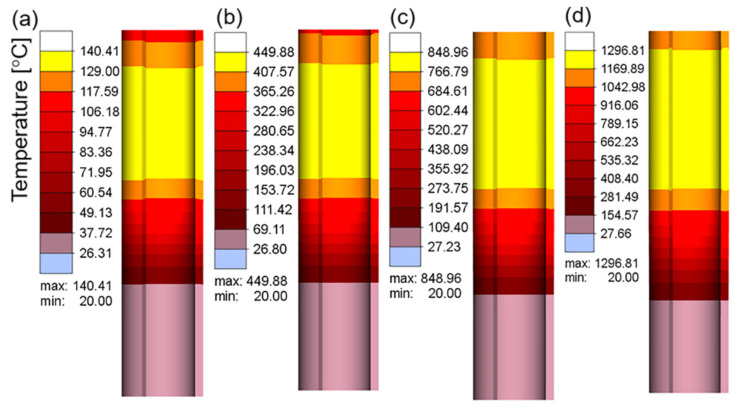
The temperature field of the workpiece at a current frequency of 10 kHz and different current values: (**a**) 100 A; (**b**) 200 A; (**c**) 300 A; (**d**) 400 A.

**Figure 9 materials-16-03938-f009:**
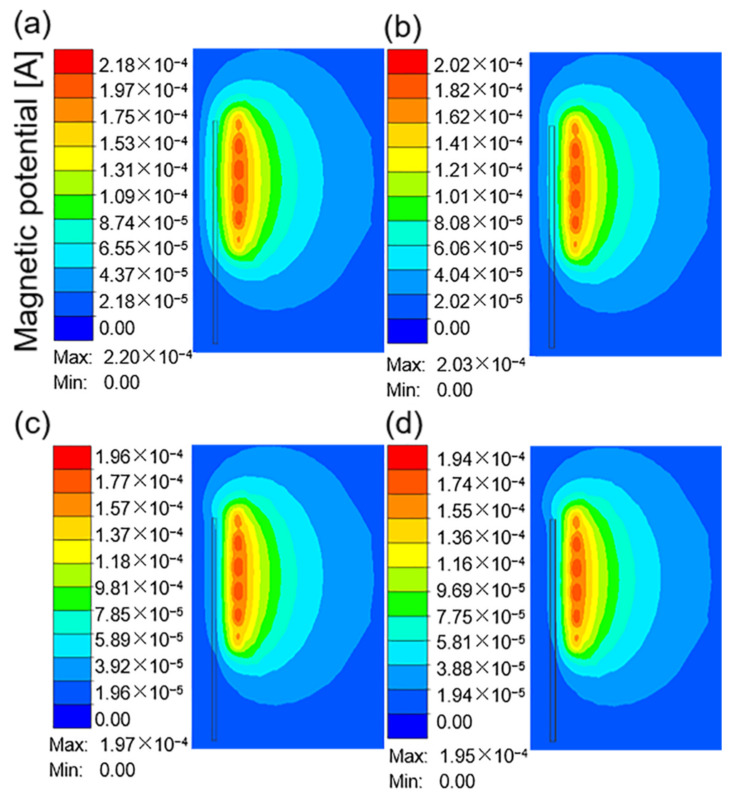
The magnetic potential around the coil at a current value of 100 A and different current frequencies: (**a**) 10 kHz; (**b**) 20 kHz; (**c**) 30 kHz; (**d**) 40 kHz.

**Figure 10 materials-16-03938-f010:**
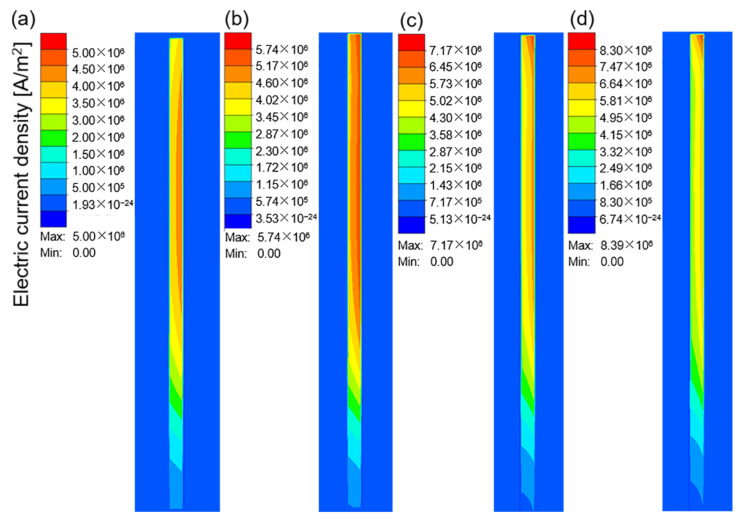
The electric current density in the workpiece at a current value of 100 A and different current frequencies: (**a**) 10 kHz; (**b**) 20 kHz; (**c**) 30 kHz; (**d**) 40 kHz.

**Figure 11 materials-16-03938-f011:**
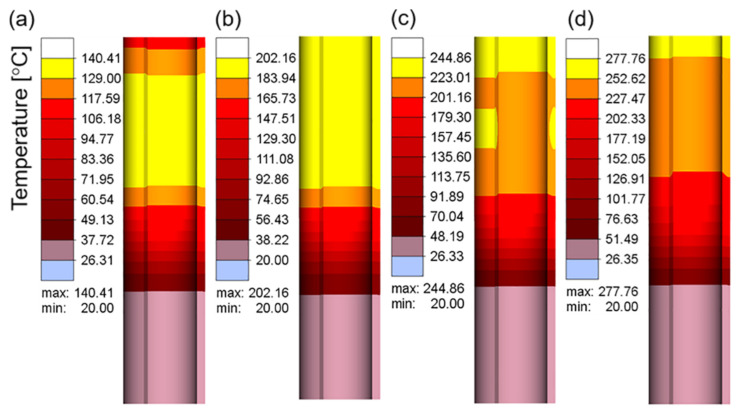
The temperature field of the workpiece at a current value of 100 A kHz and different current frequencies: (**a**) 10 kHz; (**b**) 20 kHz; (**c**) 30 kHz; (**d**) 40 kHz.

**Figure 12 materials-16-03938-f012:**
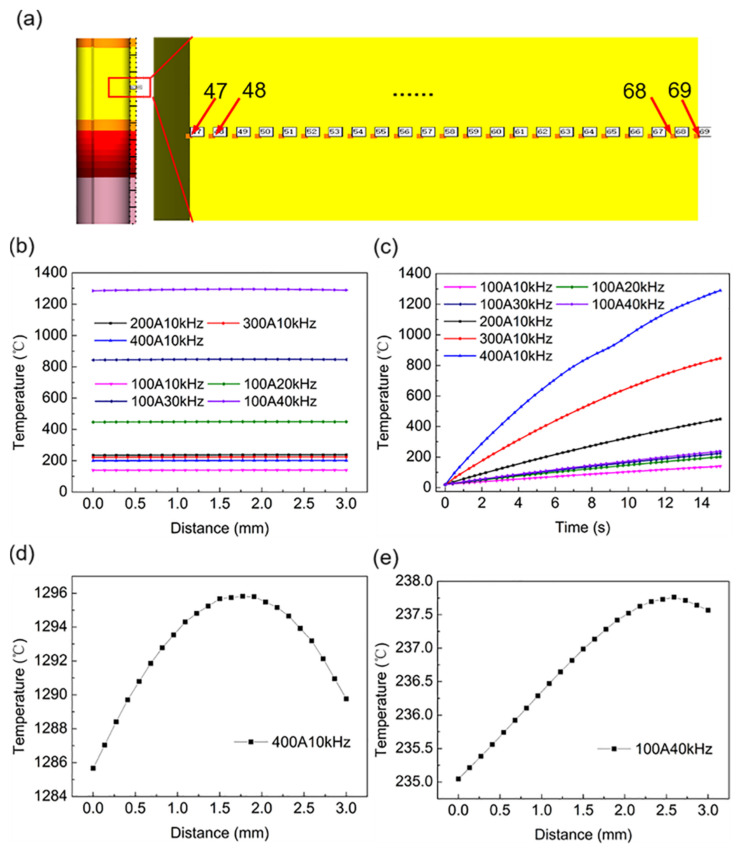
The temperature of the selected point. (**a**) Position of the selected points; (**b**) wall thickness direction temperature (47–69 points); (**c**) the temperature of point 69 in 15 s; (**d**,**e**) wall thickness direction temperature (small-scale).

**Figure 13 materials-16-03938-f013:**
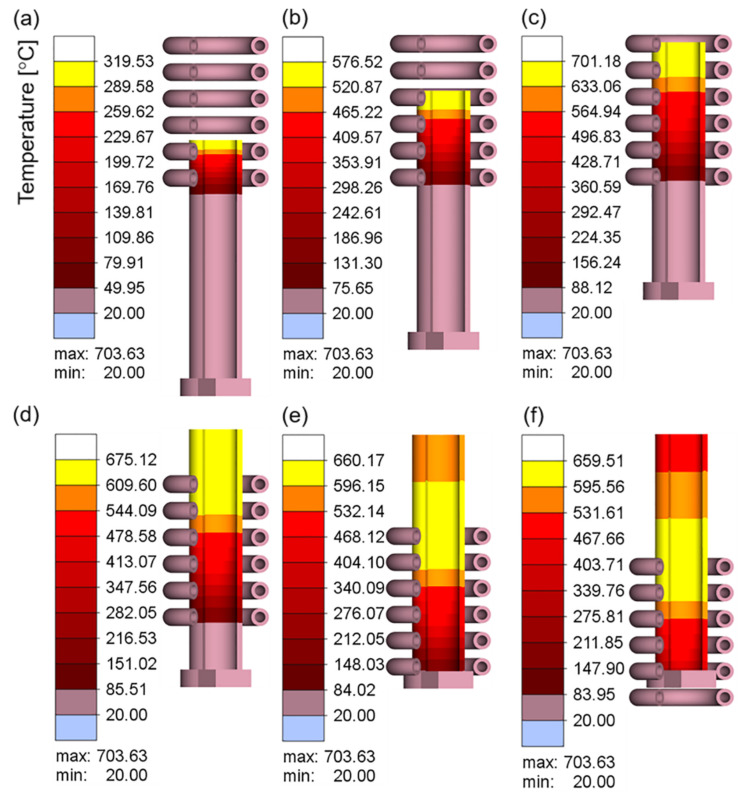
Temperature field of the workpiece during stepwise feeding for different times: (**a**) 9 s; (**b**) 18 s; (**c**) 27 s; (**d**) 37 s; (**e**) 46 s; (**f**) 52 s.

**Figure 14 materials-16-03938-f014:**
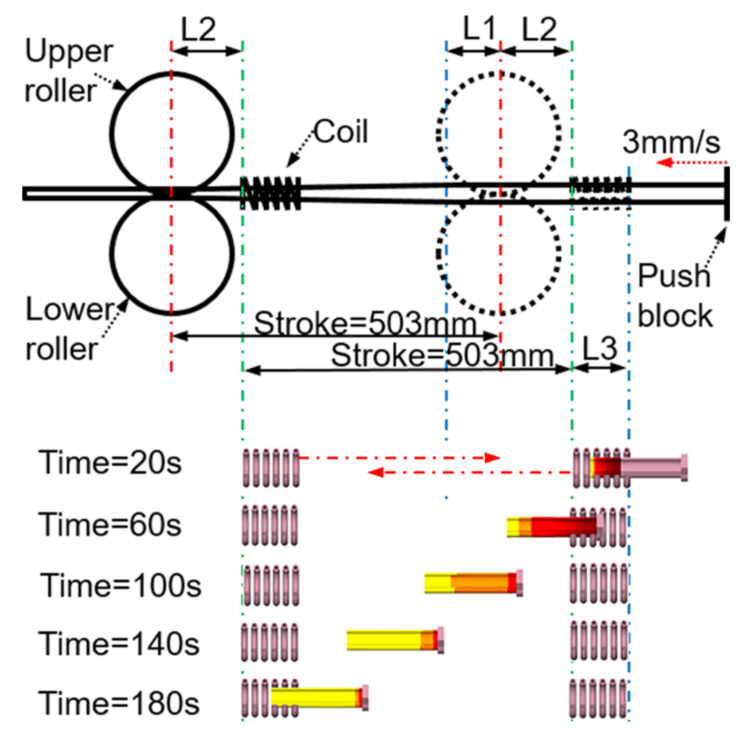
The relative position of the workpiece and the coil at different times when the superposition motion is applied to the workpiece (L1 = 90 mm, L2 = 120 mm, and L3 = 85 mm).

**Figure 15 materials-16-03938-f015:**
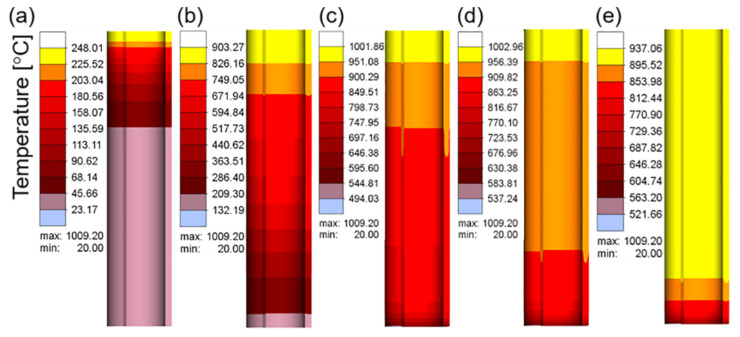
Temperature field of the workpiece during superimposed feeding at different times: (**a**) 20 s; (**b**) 60 s; (**c**) 100 s; (**d**) 140 s; (**e**) 180 s.

**Figure 16 materials-16-03938-f016:**
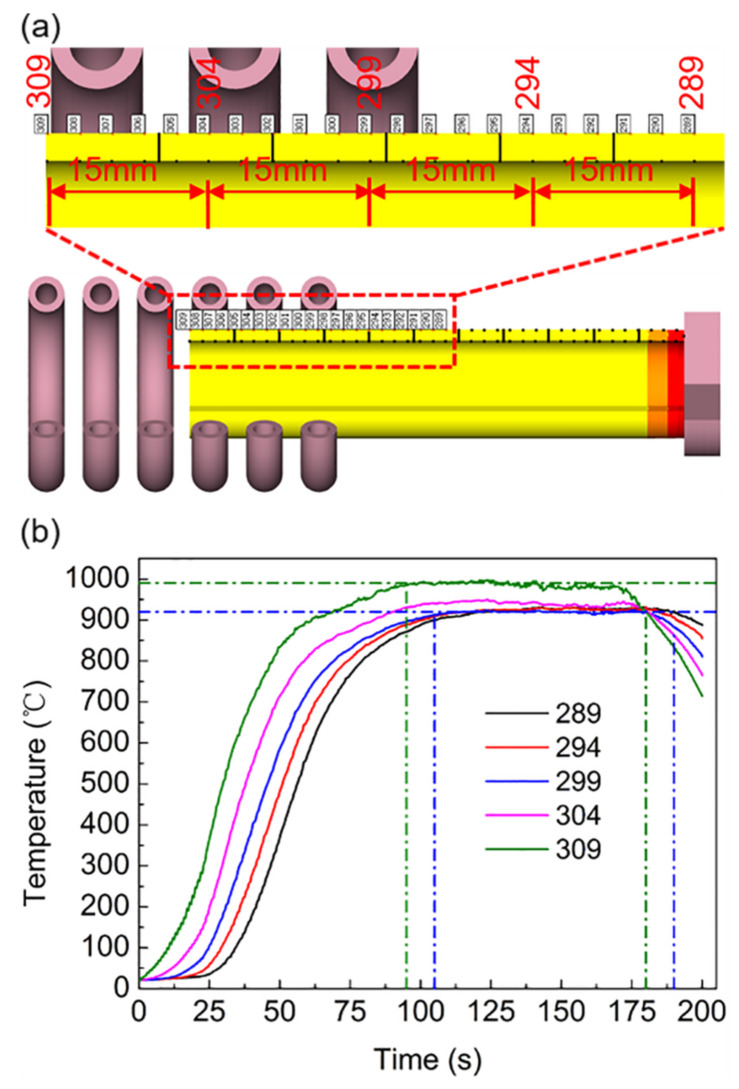
The selected point temperature–time curve of the workpiece during superimposed motion: (**a**) points location; (**b**) current value = 400 A; current frequency = 30 kHz.

**Figure 17 materials-16-03938-f017:**
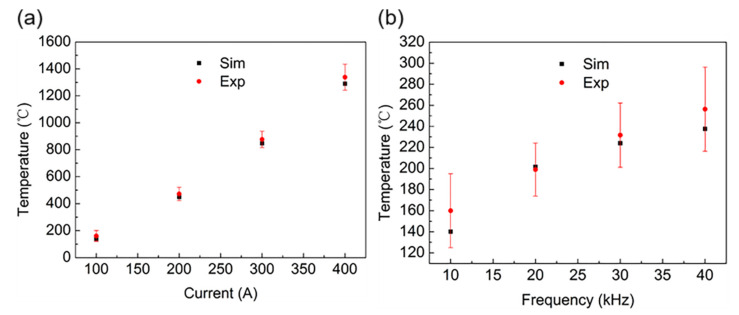
Simulation and experimental temperature at a heating time of 15 s: (**a**) f = 10 kHz, the temperature of the workpiece under various applied current values; (**b**) I = 100 A, the temperature of the workpiece under different current frequencies.

**Figure 18 materials-16-03938-f018:**
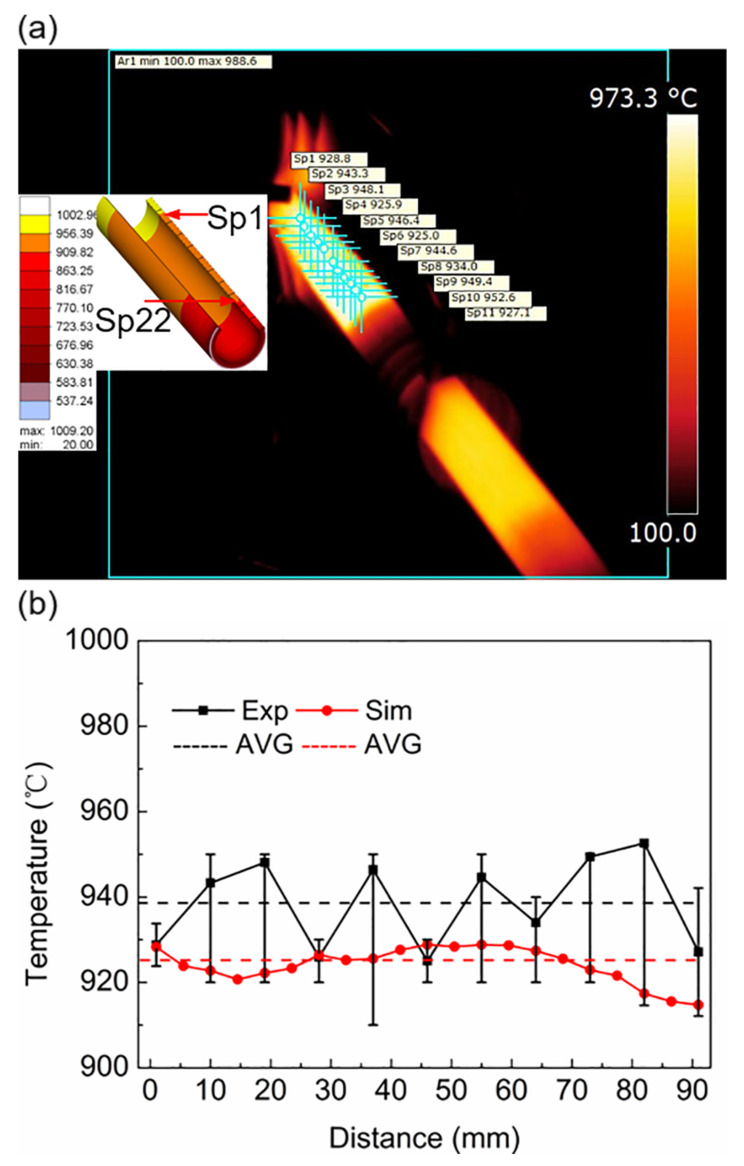
Simulation and experimental temperature of the workpiece during superimposed motion at a heating time of 140 s: (**a**) The experimental and simulated temperature fields and points location; (**b**) the selected point temperature–distance curve.

**Table 1 materials-16-03938-t001:** Mesh parameters of the induction heating parts.

Part	Element Type	Element Size (mm)	Element Count
Coil	advancing front quad (112)	0.241	3149
Tube	advancing front quad (10)	0.684	3165
Inner air	advancing front quad (112)	2.313	7539
Outer air	advancing front quad (112)	40	286
Far field	advancing front quad (112)	5	160

**Table 2 materials-16-03938-t002:** Parameters of the copper coil and air.

Physical Parameters	Copper Coil	Air
Thermal conductivity W/(m·K)	100	0.0262
Heat capacity J/(g·K)	0.385	1.005
Loss factor	0.9	0.9
Resistivity Ω·m	1.7 × 10^−8^	1.0 × 10^20^
Electrical conductivity 1/(Ω·m)	1.3 × 10^16^	1.0 × 10^−20^
Relative dielectric constant	18.1	1.00059
Relative permeability	0.999994	1.000004

**Table 3 materials-16-03938-t003:** Nomenclature.

Parameter	Significance	Unit
→	Vector	
∇	Nabla	
E	Electric field intensity	V/m
D	Electric flux density	C/m^2^
H	Magnetic field intensity	A/m
B	Magnetic flux density	T
J	Electric current density	A/m^2^
λ	Thermal conductivity	W/m·K
h	Convection heat transfer coefficient	W/(m^2^·K)
ρ	Charge density	C/m^3^
$\rho^{\prime }$	Density of workpiece	kg/m^3^
c	Specific heat capacity	J/(kg·K)
ε	Emissivity	
$\sigma_{\mathrm{sb}}$	Stefan–Boltzmann constant	
T	Temperature	K
Q	Heat source	W/m^3^
σ	Electrical conductivity	1/(Ω⋅m)

## Data Availability

Not applicable.

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
