# Peer review of "Numerical Study of Ti6Al4V Alloy Tube Heated by Super-Frequency Induction Heating"

_materials, 2023, doi:10.3390/ma16113938_

Round 1

Reviewer 1 Report

Paper presents quite interesting approach to a practical problem of induction heating process of titanium pipes.  The numerical analysis and experimental validation were provided. However more detailed description of the model is missing. Authors could neglect part of the paper with general, well  equations (1-5) and boundary conditions (6-7). Instead of that it would be better to describe precisely  numerical model, applied simplifications, error analysis  etc. Additionally equations are written not correctly (look IEC or IEEE standards!). Table 3 contains strange data. For instance Why to write unit of electric field intensity as N/C instead of  V/m  ? Unit of Volume charge density is not C/m3. It is C/m3

Author Response

Dear reviewer:

We feel great thanks for your professional review work on our article. (Our article titled "Numerical study of Ti6Al4V alloy tube heated by super-frequency induction heating" Manuscript ID: materials-2409906). As you are concerned, there are several problems that need to be addressed. According to your nice suggestions, we have made extensive corrections to our previous draft, changes/additions to the manuscript are marked up using the “Track Changes” function of Word, such that changes can be easily viewed. We hope that this revised version of the manuscript will meet your expectations. The detailed corrections are listed below.

  1. We extensively rewrite and correct lines 109 to 136 in Section 2.1, and add descriptions of model simplification and mesh convergence.
  2. In section 2.2, lines 183 to 212 were simplified and modified. Additionally the equations were corrected.
  3. Table 3 was corrected according to the rewritten content.

Kind regards,

Cheng Liu

Reviewer 2 Report

Dear authors!

Please, give more details for fig4 b and c? What is difference?

Also you say about fig5 one times, but figer has 4 pictures. Discribe all of them, please. Fig 2, 3, 6, 7, 8, 9, 10, 11, 13, 15 also has a lot pictures and onlu one comment in text.

Best regards, reviewer

Dear Editor,

this paper is original and interessting for this field.

After adding more details aboul all pictures, I think, it would de better to accept it.

Best regards, Dr. Antonina Karlina

Author Response

Dear reviewer:

We feel great thanks for your professional review work on our article. (Our article titled "Numerical study of Ti6Al4V alloy tube heated by super-frequency induction heating" Manuscript ID: materials-2409906). As you are concerned, there are several problems that need to be addressed. According to your nice suggestions, we have made extensive corrections to our previous draft, changes/additions to the manuscript are marked up using the “Track Changes” function of Word, such that changes can be easily viewed. We hope that this revised version of the manuscript will meet your expectations. The detailed corrections are listed below.

  1. We deleted Figure 4 and replaced it with a new Figure 4, which increased the abscissa to 10 seconds to show the difference between Figure 4b and c more clearly. Two auxiliary dashed lines are added to show that the vertices of the ordinate of Figure 4c are gradually rising, while Figure 4b is unchanged.
  2. The text comments of Figure 2a and b are added in lines 113 and 116. The comments of Figure 3a and b are added in line 129. The text comments of Figure 6a, b, c and d are added in line 248. The text comments of Figure 7a, b, c and d are added in line 259. The text comments of Figure 8a, b, c and d are added in line 265. The text comments of Figure 9a, b, c and d are added in line 282. The text comments of Figure 10a, b, c and d are added in line 290. The text comments of Figure 11a, b, c and d are added in line 295. The text comments of Figure 13a, b, c, d, e and f are added in line 345. The text comments of Figure 15a, b, c, d and e are added in line 382.

Kind regards,

Cheng Liu

Reviewer 3 Report

The work is good and deserves to be published. But first, you must make some of the following adjustments:

1-     On line 104 put a space between "SolidWorks 2018".

2-     On line 115, put a space before and after "Forming 16".

3-     Ensure that all symbols are defined throughout the manuscript.

4-     In Table 3, should edit and check the Units column and put the exponent in the correct form.

5-     The conclusions are too long. It should be reformulated and reduced.

Minor editing of English language required

Author Response

Dear reviewer:

We feel great thanks for your professional review work on our article. (Our article titled "Numerical study of Ti6Al4V alloy tube heated by super-frequency induction heating" Manuscript ID: materials-2409906). As you are concerned, there are several problems that need to be addressed. According to your nice suggestions, we have made extensive corrections to our previous draft, changes/additions to the manuscript are marked up using the “Track Changes” function of Word, such that changes can be easily viewed. We hope that this revised version of the manuscript will meet your expectations. The detailed corrections are listed below.

  1. Added space between ' SolidWorks 2018 ' in line 152.
  2. On line 171 we checked the spaces between ' Simufact.Forming 16 ', and on line 183 ' Simufact.Forming 16 ' has been deleted due to rewriting.
  3. We checked the definition of symbols in the manuscript.
  4. The symbols and units in table 3 have been corrected.
  5. We have simplified and modified the conclusion.

Kind regards,

Cheng Liu

Reviewer 4 Report

The manuscript presents numerical simulation to analyzing the thermal and electromagnetic fields of Ti6Al4V alloys. A temperature controlled process was developed based on cold rolling mill which has been applied to verify the numerical model. In the opinion of this reviewer, the contribution could be a useful addition. The manuscript is recommended for the publication considering the following comments.

1-      In terms of numerical simulations what are the challenges of studying small and medium sized Ti6Al4V,  please discuss.

2-      According to the meh parameters in Fig.3 and Table 1,  it needs to be made clear how this meshing was adopted and meshed in the different layers. Are the results sensitive to the mesh? It is recommended to carry out a mesh sensitivity in order to find the converged mesh.

3-      Please check the values in Table 2.

4-     In the analysis, the material properties are assumed deterministic ignoring uncertainties. The effect of uncertainties can be shown in the insignificant discrepancies in the experimental and simulated temperature of figure 18.

Minor editing of English language required

Author Response

Dear reviewer:

We feel great thanks for your professional review work on our article. (Our article titled "Numerical study of Ti6Al4V alloy tube heated by super-frequency induction heating" Manuscript ID: materials-2409906). As you are concerned, there are several problems that need to be addressed. According to your nice suggestions, we have made extensive corrections to our previous draft, changes/additions to the manuscript are marked up using the “Track Changes” function of Word, such that changes can be easily viewed. We hope that this revised version of the manuscript will meet your expectations. The detailed corrections are listed below.

  1. The description and discussion of the challenges faced by small and medium-sized Ti6Al4V numerical simulation are added in section 1, lines 77-96.
  2. In section 2.1, the discussion of grid sensitivity is added from lines 122 to 136, and the original description is simplified.
  3. We have corrected Table 2.
  4. The original Figure 18 is deleted and replaced with a new Figure 18 to reflect the impact of uncertainty.

Kind regards,

Cheng Liu

Reviewer 5 Report

In the manuscript "Numerical study of Ti6Al4V alloy tube heated by super-frequency induction heating" a numerical simulation and experimental study on the ultrasonic induction heating process of a Ti6Al4V titanium-alloy tube were conducted to obtain stable heating. A finite element model of electromagnetic–thermal coupling was developed. The electromagnetic and temperature fields at different current values and current frequencies were evaluated, and the temperature field under different motion states was analyzed. A reciprocating motion online heating method was proposed.
The comments are as follows:
1. In the abstract and in the introduction, the term "narrow technological window" is used. A more detailed description of what it consists of is required.
2. The inscriptions in figure 1 and 4 are very small.
3. The resistance is the reciprocal of conductivity and it is not clear why both dependencies should be placed in Figure 5. Thermophysical properties have been calculated, but require verification and discussion, especially the increase in conductivity with increasing temperature.
4. In the Table 4 it is better to give the dimensions used, for example, the electric field strength is more often expressed in V/m. The top injections in the dimensions also need to be corrected.
5. More detailed discussions are needed about the rise in temperature of the pipe at the end when it is taken out of the coil (Figure 13f).
6. It would be interesting to know what degree of deformation has been achieved experimentally and how much this technology is superior to those used.

Minor editing of English language required.

Author Response

Dear reviewer:

We feel great thanks for your professional review work on our article. (Our article titled "Numerical study of Ti6Al4V alloy tube heated by super-frequency induction heating" Manuscript ID: materials-2409906). As you are concerned, there are several problems that need to be addressed. According to your nice suggestions, we have made extensive corrections to our previous draft, changes/additions to the manuscript are marked up using the “Track Changes” function of Word, such that changes can be easily viewed. We hope that this revised version of the manuscript will meet your expectations. The detailed corrections are listed below.

  1. In section 1, lines 37 to 45, the description and discussion of ' narrow technical window ' are added.
  2. The original Figure 1 and Figure 4 are deleted and replaced with a new picture to make the inscriptions more clear.
  3. The original Figure 5 is deleted and replaced with the new Figure 5, and only one of the two parameters of resistivity and conductivity is retained. In section 2.1, lines 165 to 171, the discussion of the reliability of the calculated thermal physical parameters is added and the relevant literature is cited.
  4. There is no 'table 4' in this article. We think the 'table 4' may be mean 'table 3', and table 3 has been corrected.
  5. Some discussions are added from line 361 to line 364, and it is pointed out that the temperature of the two ends of the billet is slightly different from that of the middle section of the billet when entering and leaving the coil. In the description of Fig.16 (from line 392 to line 401), the temperature curve of the point at the top of the tube and the subsequent part specifically describes the reason why the temperature is slightly higher when the top of the tube enters the coil. Although Fig.13 and Fig.16 are different in motion, the basic principles are the same. Therefore, this part have not given a detailed description of the temperature at both ends when the tube enters and leaves the coil. This part is only used to draw out the phenomenon that the temperature of the tube blank will decrease greatly after the tube is out of the coil, and to draw out the discussion of changing the motion mode.
  6. Some new discussions were added in lines 466 to 474 to describe the advantages of this technology in ' deformation degree ' compared to ' multi-pass cold rolling plus multiple intermediate annealing process '.

Kind regards,

Cheng Liu
